# WHEN THREE EXPERIMENTS ARE BETTER THAN TWO: AVOIDING INTRACTABLE CORRELATED ALEATORIC UNCERTAINTY BY LEVERAGING A NOVEL BIAS–VARIANCE TRADEOFF

## ABSTRACT

Real-world experimental scenarios are characterized by the presence of heteroskedastic aleatoric uncertainty, and this uncertainty can be correlated in batched settings. The bias–variance tradeoff can be used to write the expected mean squared error between a model distribution and a ground-truth random variable as the sum of an epistemic uncertainty term, the bias squared, and an aleatoric uncertainty term. We leverage this relationship to propose novel active learning strategies that directly reduce the bias between experimental rounds, considering model systems both with and without noise. Finally, we investigate methods to leverage historical data in a quadratic manner through the use of a novel cobias–covariance relationship, which naturally proposes a mechanism for batching through an eigendecomposition strategy. When our difference-based method leveraging the cobias–covariance relationship is utilized in a batched setting (with a quadratic estimator), we outperform a number of canonical methods including BALD and Least Confidence.

## 1 INTRODUCTION

In real-world scenarios where data acquisition is costly, Active Learning (AL) attempts efficient labeling of informative data points to maximize model performance (Ren et al., 2021; Settles, 2009). However, especially within the life sciences, experimental data are intrinsically noisy — commonly referred to as "aleatoric uncertainty" (Der Kiureghian & Ditlevsen, 2009). Replicates are performed to ascertain that results do not originate from biological or technical factors. Recently, there has been much interest in 'lab-in-the-loop' systems within drug discovery (Taylor-King et al., 2024) wherein a deep learning system directs wet lab experiments to achieve some goal of interest: from the identification of novel synergistic drug pairs (Bertin et al., 2023), to the prediction of transcriptomic profiles within "perturb-seq" experiments (Kovačević et al., 2025; Peidli et al., 2024). Both of the aforementioned systems, along with many others, exhibit aleatoric uncertainty. However, this uncertainty is heteroskedastic: *certain experiments are more predictable than others*. Moreover, in many situations, observations are naturally batched, meaning that within any one particular batch there is a shared noise structure, and therefore any batch selection mechanism should intelligently take this into account; see single-cell technologies for examples of this in practice (De Jonghe et al., 2024b;a), or consider how groups of experiments may share common characteristics, for example, using the same incubator. Therefore, we wish to intelligently perform replicates within said 'lab-in-the-loop' system for economical understanding of the underlying system accounting for these key features (heteroskedasticity, correlated noise within batches).

AL-style problems have historically appeared in many forms depending on whether the goal is to maximize a reward function (Sequential Model Optimization (Schagen, 1984), Bayesian Optimization (Jones et al., 1998)) or to fit a statistical or machine learning model (Bayesian Optimal Experimental Design (Lindley, 1956)). Traditional methods in AL range from heuristics, such as the least confidence (LC) strategy, where points are labeled for which a model is the most uncertain, to more sophisticated approaches such as query-by-committee, where points are labelled for which an ensemble of models most disagree with each other (Seung et al., 1992; Scherer et al., 2022), indicating regions of high epistemic uncertainty. Epistemic uncertainty is the "model uncertainty" and can be quantified as

the reduction in uncertainty through the acquisition of more data, whereas aleatoric uncertainty, inherent to the observation process, as noted, remains even with infinite data (Kendall & Gal, 2017). This distinction is crucial in AL, where the objective is to mitigate uncertainty in model predictions by strategically labeling new data points. From a Bayesian perspective, this aligns with using the expected information gain in Bayesian models or deep ensembles (Smith & Gal, 2018). Recently, Kirsch & Gal (2022) showed the connection between various AL methods and information-theoretic quantities.

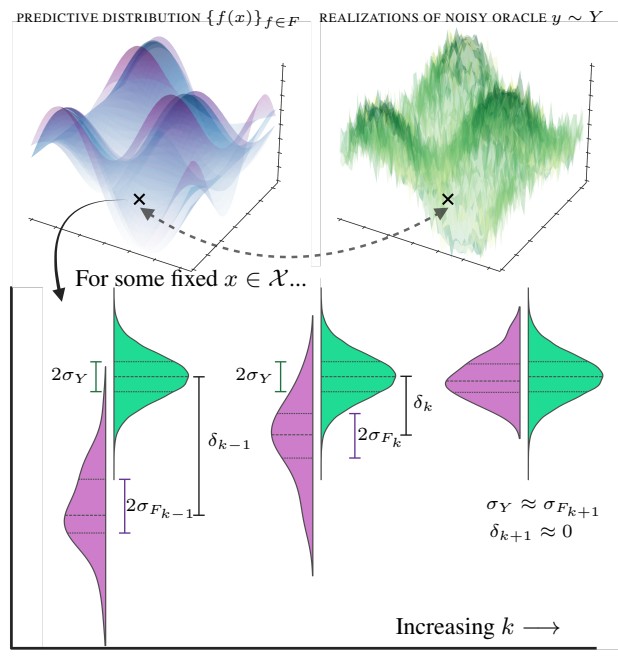

Figure 1: For some point in the state space ($x \in \mathcal{X}$), we have a (known) predictive distribution ($\{f(x)\}_{f \in F}$) and a noisy oracle that admits realizations ($y \sim Y$) with the underlying distribution allowed to vary as a function of the state space (i.e., $Y = Y(x)$).

Our task is twofold: first to match the expected value of $Y$ (the random variable representing the ground truth process) with the expected value of $F$ (the random variable for the distribution of fitted functions), i.e., the bias tends to zero; second we wish to obtain robust estimates, which can be achieved by having the distributions approximately match.

In this work, we note that for regression problems the bias–variance tradeoff can be applied and the expected mean squared error (EMSE) can be interpreted as the sum of an epistemic uncertainty term, the bias squared, and an aleatoric uncertainty term; more general bias–variance tradeoffs exist through the use of Bregman divergences (Pfau, 2013; Adlam et al., 2022). Consider Figure 1, one would ideally like to select points in the state space such that the bias is close to zero, but also that the predictive distribution approximately matches the underlying noisy oracle — we do not want our predictions being more or less certain than the underlying truth. Naturally, when considering regions in the state space to select, some will correspond to areas whereby the bias or epistemic uncertainty rapidly collapses — these should be prioritized for labeling. We achieve this through calculating an approximation to the derivative of the EMSE between experimental rounds — leading to our paper title: two rounds of experiments can be used to estimate the gradient of the EMSE, which is then exploited in a third round of experiments (or indeed, any future experiments). This approach essentially requires us to estimate the EMSE at unobserved points in the state space; generally, this is a challenging thing to do. However, we note that the EMSE is a squared $L^2$ norm in a Hilbert space, and we can therefore use the associated inner product to recast the problem to leverage a novel "cobias–covariance" tradeoff to leverage unique historically collected data points quadratically (as opposed to linearly) and further improve model accuracy. This cobias–covariance relationship also provides a natural framework to account for correlated sources of noise. Furthermore, through eigendecomposition, we have a mathematically grounded mechanism for selecting batches.

We apply our collection of methods to both problems without noise, which we refer to as "Type I problems" (i.e., standard AL), and noisy systems that can be further divided into "Type II problems" (uncorrelated noise) or "Type III problems" (with correlated noise). As there are multiple means by which one can batch queries (i.e., nominating multiple points concurrently for labeling), we consider two scenarios in which one can choose only one point at a time before the model is re-trained and also when one nominates $m \in \mathbb{N}_+$ points to label. We focus on a deliberately challenging artificial toy system with different types of noise terms added (where other AL methods fail).

We find that our suite of methods, *Avoiding Intractable Correlated Aleatoric Uncertainty* (AICAU), can outperform other methods in a range of model settings. Our approach appears to be original with a clear route forward to expand the scope, reliability, and applicability of the method, for example, via Bregman divergence formulations. Conceptually, we are posing the active learning problem in a manner that is more susceptible to the analysis of functions, as opposed to the more common approach of using Bayesian and/or information theory approaches.

## 2 METHOD CONCEPT

We first provide a mathematical description of our problem: to use a distribution of functions to learn a noisy function. For ease of reading, this work uses notation similar to a recent AL survey (Ren et al., 2021).

### 2.1 PROBLEM STATEMENT

For queried state $x$, we let $\mathcal{X}$ be the state space for each sample (discrete or continuous), and $y$ is the recorded label drawn from random variable $Y$ in space $\mathcal{Y}$. Through multiple rounds of experiments, in round $k$, queried data then takes the form $Q_k = \{(x_i, y_i)\}_{i=1}^m$ for $(x_i, y_i) \in \mathcal{X} \times \mathcal{Y}$ such that $y_i \sim Y$ is a realisation of random variable $Y$. Let $L_0$ denote the data used for pretraining, and then $L_k = Q_k \cup L_{k-1}$ as all of the labelled data up until experimental round $k$.

We write $F_k$ as a distribution of functions trained on $L_k$ whereby $f \sim F_k \in \mathcal{F}$ maps $f : \mathcal{X} \to \mathcal{Y}$. If all of the functions have the same functional form, then we can write $F_k(x) = f(x\,;\Theta)$ where $\Theta$ is a random variable over the parameter space. Regardless of the underlying model for $F_k$, we write the mean and variance as

$$\mu_{F_k}(x) := \mathbb{E}_{\mathcal{F}}[F_k(x)] \quad \text{and} \quad \sigma^2_{F_k}(x) := \mathbb{E}_{\mathcal{F}}\left\{[F_k(x) - \mu_{F_k}(x)]^2\right\}. \tag{1}$$

In the case where $f$ is a finite set (e.g., when using deep ensembles), we refer to $\{f(x)\}_{f \in F_k}$ as the *predictive distribution* for a fixed $x \in \mathcal{X}$ and we can calculate estimates of expected values in the standard manner, i.e., $\widehat{\mu_{F_k}}(x) = \sum_{f \in F_k} f(x)/|F_k|$ and sample variance analogously.

We assume that there exists a deterministic mapping from $\mathcal{X}$ to $\mathcal{Y}$, we write $\mu_Y : \mathcal{X} \to \mathcal{Y}$. However, due to the presence of experimental noise, we assume the existence of a generic aleatoric noise term represented by random variable $W = W(x)$ dependent on the region of state space being sampled: most real-world systems of interest present *heteroskedastic* measurement error dependent on the underlying state space. Therefore, the observed value pairs $(x, y) \in \mathcal{X} \times \mathcal{Y}$ obey the following relationship

$$y \sim Y(x) = \mu_Y(x) + W(x). \tag{2}$$

For simplicity and later ease of notation, we specify that

$$\mathbb{E}_{\mathcal{Y}}[W(x)] \equiv 0 \quad \text{and} \quad \mathbb{E}_{\mathcal{Y}}[W^2(x)] = \sigma^2_Y(x). \tag{3}$$

Moreover, whilst the noise pattern for $W(x)$ may be highly dependent on $x \in \mathcal{X}$, we are still able to recover $\mu_Y(x)$ as the number of samples goes to infinity. In batched settings, $W(x)$ may be correlated across the input space, discussed in Section 2.3.2.

In a perfect world, $\mathbb{E}_{\mathcal{F}}[F_{k^*}](x)$ would agree perfectly with $\mu_Y(x)$ after a small number of $k_* \in \mathbb{N}_+$ experiments guided by a sequential model optimisation strategy. At least initially, we cannot expect this to be the case. Therefore, we assume the existence of a bias term $\delta_k(x)$, which we write as

$$\delta_k(x) = \mu_{F_k}(x) - \mu_Y(x). \tag{4}$$

In the limit, one can write that $\lim_{k \to \infty} \delta_k(x) = \delta_\infty(x)$ and if the space of functions $\mathcal{F}$ is suitably flexible, then $\delta_\infty \equiv 0$. We note that for some real world systems, many deep learning models cannot capture the discontinuous nature of $\mu_Y(x)$ within the state space and therefore $\delta_\infty(x) \neq 0$.

Our goal is to devise strategies that rapidly reduce the mean squared error (MSE) over a discretized space $\mathcal{X}$ via finite vector $\vec{x} = (x_1, \ldots, x_n)$ with $x_i \in \mathcal{X}$. To evaluate the MSE, we allow access to the true $\mu_Y(x)$ to calculate the MSE as $\sum_{i=1}^n \delta_k^2(x_i)/n = \sum_{i=1}^n [\mu_{F_k}(x_i) - \mu_Y(x_i)]^2/n$, which we calculate over a discretization of $\mathcal{X}$, written $\vec{x} = (x_1, \ldots, x_n) \in \mathcal{X}^n$.

## 2.2 STATEMENT OF BIAS−VARIANCE TRADEOFF

In the following, we assume we wish to predict a real number (so $\mathcal{Y} \equiv \mathbb{R}$) and by writing the *pointwise* expected mean squared error (PEMSE) at $x \in \mathcal{X}$, denoted by $\tau_k(x)$, we state the standard bias-variance tradeoff relationship (Adlam et al., 2022) applied to our problem as

$$\tau_k(x) = \mathbb{E}_{\mathcal{F} \times \mathcal{Y}} \left\{ [F_k(x) - Y(x)]^2 \right\} = \underbrace{\sigma^2_{F_k}(x)}_{\text{Epistemic uncertainty}} + \underbrace{\delta^2_k(x)}_{\text{Bias}^2} + \underbrace{\sigma^2_Y(x)}_{\text{Aleatoric uncertainty}} . \quad (5)$$

Equation (5) holds provided the test-time label noise is independent of the fitted predictor, i.e. $\text{Cov}(F_k(x), Y(x)) = 0$. On the assumption that $\mathcal{X}$ can be discretized into a finite vector $\vec{x} = (x_1, \ldots, x_n)$ with $x_i \in \mathcal{X}$, we take the average over Equation (5) to calculate the (global) expected mean squared error (EMSE) at round $k$, which we can view as an unseen loss $\mathcal{L}_k = \sum_{i=1}^{n} \tau_k(x_i)/n$ that we are trying to reduce as $k \to \infty$. Naturally, only the epistemic uncertainty and the bias are *reducible*, and this becomes the target for our AL strategy.

## 2.3 PROBLEM VARIATIONS

### 2.3.1 TYPE I PROBLEM: NOISELESS SYSTEMS

In systems without aleatoric noise, that is, $W \equiv 0$, every measurement of $y$ is exact, and therefore there is no utility in evaluating $x$ more than once. Therefore, we enforce that there are no repeat measurements and therefore $Q_k \cap L_k = \emptyset$. The goal is therefore to predict $y = \mu_Y(x)$ for unseen points. The number of points shrinks each iteration and therefore the ability for a model to learn quickly is of paramount importance.

In a world without aleatoric uncertainty (i.e., $\sigma^2_Y \equiv 0$), we wish to reduce the bias as fast as possible whilst accounting for variability in the predictive distribution. Considering that the EMSE can be written

$$\mathcal{L}_k = \frac{1}{n} \sum_{i=1}^{n} \tau_k(x_i) = \frac{1}{n} \sum_{i=1}^{n} \left[ \sigma^2_{F_k}(x_i) + \delta^2_k(x_i) \right] \quad (6)$$

then a number of possible acquisition functions are reasonable, e.g., aim to reduce the bias term in (6). However, we do not know $\delta^2_k(x)$ for all $x \in \mathcal{X}$, so it must be estimated using another method, e.g., via a neural network, or even interpolation. Assuming an approximation can be found, we consider the acquisition functions in Table 1.

| BASE METHOD | ACQ. FUNC. $\alpha_k(x)$ | 'DIFFERENCE' ACQ. FUNC. |
|---|---|---|
| RANDOM | CONSTANT | N/A |
| LEAST CONFIDENCE | $\sigma^2_{F_k}(x)$ | $\kappa(\sigma^2_{F_k}(x))$ |
| BIAS REDUCTION | $\delta^2_k(x)$ | $\kappa(\delta^2_k(x))$ |
| PEMSE | $\sigma^2_{F_k}(x) + \delta^2_k(x)$ | $\kappa(\tau_k(x))$ |

Table 1: Acquisition functions proposed in this article.

### 2.3.2 TYPE II/III PROBLEM: NOISY SYSTEMS

In systems with aleatoric noise, then there may be benefit to evaluating the same data point $x \in \mathcal{X}$ multiple times to obtain multiple realisations of $y$. Whilst we obtain values of $y \sim Y = \mu_Y(x) + W(x)$, we compare algorithms on the ability to learn $y = \mu_Y(x)$ across all points. Finally, we also separate between systems with *uncorrelated* noise (Type II) when $\mathbb{E}_{\mathcal{Y}}[W(x)W(x^*)] = 0$ and *correlated* noise (Type III) when $\mathbb{E}_{\mathcal{Y}}[W(x)W(x^*)] = \rho(x, x^*)$ for $\rho(x, x^*) \neq 0$ if $x \neq x^*$, see Table 2 for a summary.

Consider an active learning strategy in the presence of aleatoric uncertainty, what kind of properties would it have? Across multiple rounds of active learning, one would imagine that: a.) regions of $\mathcal{X}$ where $\tau_k$ rapidly decreases between rounds are areas of high absolute bias or epistemic uncertainty;

and b.) regions of $\mathcal{X}$ where $\tau_k$ only minimally decreases between rounds are areas of high (intractable) aleatoric uncertainty. Pertinent to (a.), to identify regions of $\mathcal{X}$ of interest, our acquisition function considers an approximation to the negative gradient of the PEMSE, more specifically

$$-\frac{\partial}{\partial k} \tau_k \approx \tau_{k-1} - \tau_k \tag{7}$$

is positive in areas of rapidly decreasing bias or epistemic uncertainty. Naturally, $k$ is not a continuous variable, however it may be useful to think in this manner as future work could consider the use of advanced numerical schemes to achieve robust estimates of this gradient. For ease of exposition, we define the difference operator

$$\kappa[g_k](x) := g_{k-1}(x) - g_k(x). \tag{8}$$

If we wish to then consider how the EMSE decreases from one round to another, consider

$$\kappa(\mathcal{L}_k) = \frac{1}{n} \sum_{i=1}^{n} \left[ \tau_{k-1}(x_i) - \tau_k(x_i) \right] = \frac{1}{n} \sum_{i=1}^{n} \left[ \kappa(\sigma_{F_k}^2(x_i)) + \kappa(\delta_k^2(x_i)) \right] \tag{9}$$

because the aleatoric error term $\sigma_Y^2(x)$ in equation (5) cancels. These observations motivate the 'difference' acquisition functions in Table 1. For completeness, we also consider the reducible component of the PEMSE as the corresponding non-difference strategy, i.e., as already described in Equation (6) — even through an aleatoric term is present in Type II/III problems that we ignore.

| TYPE | ALEATORIC NOISE | GENERAL FUNCTION STRUCTURE | SPECIFIC TOY MODEL |
|------|-----------------|----------------------------|--------------------|
| I | NONE | $y = \mu_Y(x)$ | $\mu_Y(x) = \sin\left(\frac{3x_1}{2}\right) \sin\left(\frac{3x_2}{2}\right)$ |
| II | UNCORRELATED | $y \sim Y = \mu_Y(x) + W(x)$ 
 $\mathbb{E}\left[W(x)W(x^*)\right] = 0$ 
 FOR $(x \neq x^*)$ | $\mu_Y(x)$ AS TYPE I, 
 $W(x) = \varepsilon \sqrt{1 - \mu_Y(x)^2}/10$ 
 $\varepsilon \sim \mathcal{N}(0, 1)$ |
| III | CORRELATED | $\vec{y} \sim \vec{Y} = \vec{\mu}_Y(\vec{x}) + \vec{W}(\vec{x})$ 
 $\mathbb{E}\left[W(x)W(x^*)\right] = \rho(x, x^*)$ | $[\vec{\mu}_Y(\vec{x})]_i$ AS TYPE I, 
 $[\vec{W}(\vec{x})]_i = \varepsilon(x_i) \sqrt{1 - \mu_Y(x_i)^2}/10$ 
 $\vec{\varepsilon} \sim \mathcal{N}(\vec{0}, \Sigma)$ 
 $\rho(x, x^*) = \exp\{-2\|x - x^*\|_2/\pi\}$ |

Table 2: Categorization of types of aleatoric noise in active learning and toy problem investigated in this paper.

### 2.4 ACQUISITION FUNCTION SELECTION WITH PERFECT INFORMATION (CHEATING!)

We would like to understand the relative performances of the strategies proposed in Table 1 using a well-understood, but very challenging, toy system. Both Bias Reduction (BR) and PEMSE methods require the estimation of the bias in Equation (5). To understand the rate of improvement in the MSE without errors associated to the approximation, we allow for perfect information, i.e., all methods have access to the true distribution $Y$ across state space $\mathcal{X}$. We also compare to a standard method popular in the literature, BALD (Houlsby et al., 2011).

We are interested in toy systems with a number of desired properties. It should be simple to visualise and understand how the active learning strategy has selected points for labelling. The presence of heteroskedastic aleatoric noise to selectively obscures signal in specific regions of the state space, such that random equidistributed sampling is suboptimal and more sophisticated approaches can be meaningfully benchmarked against one another. With Type III problems in mind, noise can be correlated across the state space such that realisations are intrinsically batched.

To fulfil all of the above properties, we focus on a 2-dimensional toy system described in Table 2, i.e., $x = (x_1, x_2)$ in $\mathcal{X} = [0, 2\pi]^2$ and recorded labels are real numbers ($y \in \mathbb{R}$). Conceptually, $\mu_Y(x)$ is a function bounded between $-1$ and $+1$, and when at either boundary $\sigma_Y^2(x)$ is small; conversely when $\mu_Y(x)$ is close to 0, then $\sigma_Y^2(x)$ is comparatively large. For the Type I problem, we

set $W(x) \equiv 0$, and for Type III problems the noise term $\varepsilon$ is correlated across $\mathcal{X}$, see Table 2 for a summary, and Appendix B for further details on the numerical method.

To benchmark how well we are able to learn $\mu_Y(x)$, we plot our MSE for all 3 problems in Figure 2 either with 10 (main text) or 100 initial points (Appendix A). We clearly see that in all 3 scenarios, any method that exploits the bias is clearly superior to LC, BALD, or random selection. We hypothesize that the lack of clear benefit using the difference-based methods is because we do not see large changes in either the bias or the PEMSE between two consecutive rounds (only one point was selected per round). Therefore, the benefits should become apparent in a batched setting, demonstrated in Section 4.2.

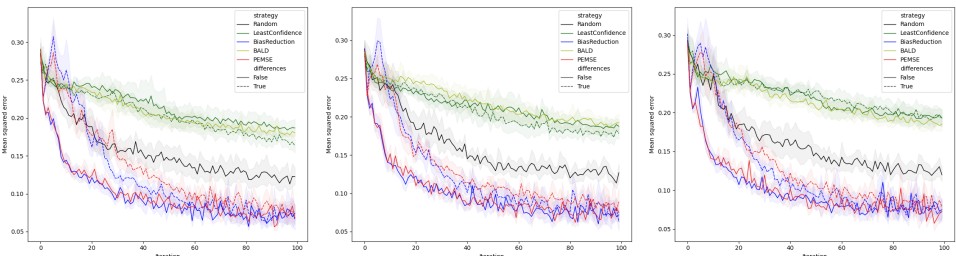

Figure 2: Assessment of acquisition functions proposed in Table 1 with perfect information available. We show all three problems in Table 2: Type I (left), Type II (middle), and Type III (right).

Whilst we have shown the benefit to using bias-based approaches *in theory*, we are now stuck with two key problems before we can apply this *in practice*: (1.) how do we robustly estimate the bias?; and (2.) how do we construct diverse batches of points to be selected together? Both of these problems are considered in Section 3.

## 3  METHOD IN PRACTICE

From Equation (5), all acquisition functions can be calculated if one has an estimator for the bias, therefore we focus our efforts here. Hypothetically, one could use another model to do this via *direct estimation* (e.g., using a Gaussian process), however a more sophisticated and potentially more numerically stable approach uses *quadratic estimation* that we detail below by leveraging a novel "cobias–covariance" tradeoff. We summarize the complete workflow in Figure 3.

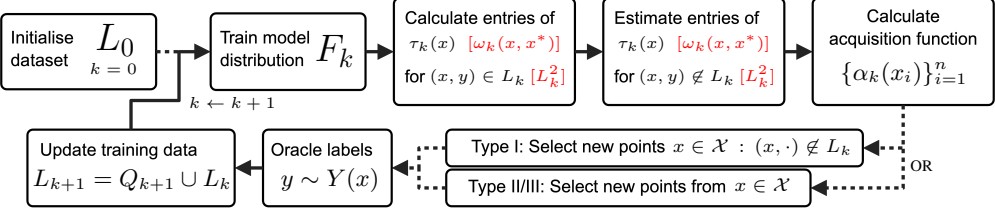

Figure 3: Diagram of active learning procedure. Alternative cobias–covariance calculation in red. In practice, estimation of the (co)bias is more stable than working with $\tau_k$ or $\omega_k$ directly, see Appendix E for calculation details.

### 3.1  A NOVEL COBIAS–COVARIANCE TRADEOFF

We note that the MSE is the squared $L^2$ norm of the prediction error. Let $\langle u, v \rangle_{L^2} := \mathbb{E}_{\mathcal{F} \times \mathcal{Y}}[\, u\, v\,]$ denote the $L^2$ inner product averaging over both model randomness and measurement noise, with induced norm $\|u\|_{L^2}^2 = \langle u, u \rangle$. Then, for a fixed input $x$,

$$\tau_k(x) = \|F_k(x) - Y(x)\|_{L^2}^2 = \langle F_k(x) - Y(x),\, F_k(x) - Y(x) \rangle_{L^2}. \tag{10}$$

In which case, we consider adapting Equation (5) but for non-identical elements of $\mathcal{X}$ to derive a cobias–covariance tradeoff

$$\omega_k(x, x^*) = \mathbb{E}_{\mathcal{F} \times \mathcal{Y}} \left\{ [F_k(x) - Y(x)][F_k(x^*) - Y(x^*)] \right\} = \sigma_{F_k}(x, x^*) + \delta_k(x)\delta_k(x^*) + \sigma_Y(x, x^*), \tag{11}$$

where $(x, x^*) \in \mathcal{X} \times \mathcal{X}$, see Appendix C for a derivation. With $\mathcal{X}$ discretized into a finite vector $\vec{x} = (x_1, \ldots, x_n)$ for $x_i \in \mathcal{X}$, then we can write Equation (11) in matrix form for $\Omega_{ij}^{(k)} = (\omega_k(x_i, x_j))_{ij}$ and

$$\Omega^{(k)} = \Sigma_{F_k} + \Delta_k + \Sigma_Y \,, \tag{12}$$

for covariance matrices $\Sigma_{F_k}$, $\Sigma_Y$ and rank-1 cobias matrix $\Delta_k = \vec{\delta}_k \vec{\delta}_k^\mathsf{T}$. From $\Delta_k$ one can recover $\vec{\delta}_k$ up to a global sign via any rank-1 factorization (e.g., the top eigenvector scaled appropriately); the diagonal alone determines only the magnitudes $|\delta_k(x_i)|$. For Type I problems $\Sigma_Y \equiv 0$; for Type II problems $\Sigma_Y$ is a diagonal matrix; and for Type III problems $\Sigma_Y$ is symmetric with non-diagonal elements.

## 3.2 QUADRATIC BIAS ESTIMATION

As explained in Section 2, we need to estimate the bias. For points that we have seen historically, we can precalculate elements of $\Delta_k$, but we will have missing entries corresponding to rows and columns corresponding to points $x \in \mathcal{X}$ that have not been seen before. Therefore, we need to predict missing entries via an estimator $Q : \mathcal{X} \times \mathcal{X} \to \mathbb{R}$. There are a number of approaches to this task, including symmetric matrix completion problems, whereby $(x, x^*) \in \mathcal{X} \times \mathcal{X}$ could be used as "side information" (Xu et al., 2013). Because $\Delta_k = \vec{\delta}_k \vec{\delta}_k^\mathsf{T}$, we are motivated to exploit the low rank structure of the matrix. We use a symmetric neural network for matrix completion (written $Q$) that also leverages the $(x, x^*)$ information, we write

$$Q(x, x^*) = \psi(x)^\mathsf{T} \psi(x^*) \equiv Q(x^*, x) \,, \tag{13}$$

where $\psi : \mathcal{X} \to \mathbb{R}^h$ is a neural network that maps to hidden dimension of size $h$. When using a neural network formulation, in order to avoid double counting off-diagonal entries, we restrict the training data to the lower triangle of symmetric matrix $\Omega^{(k)}$ (analogously, one could use the upper triangle).

With the previous use of the bias–variance tradeoff in Section 2.2, if we have observed $l_k$ unique $x \in \mathcal{X}$ in round $k$ (for a Type I problem, $l_k = |L_k|$), these points become our training data to infer $\tau_k(x)$ for all $x \in \mathcal{X}$. In this improved cobias–covariance formulation, we now have $l_k(l_k - 1)/2$ unique points to train from.

The benefits of *quadratic estimation* relate exclusively to scenarios when one wishes to leverage off-diagonal entries of $\Delta_k$, see Section 3.3. In *direct estimation*, to calculate $\Delta_k$ we estimate missing values of $\vec{\delta}_k$ and build the prediction vector $\vec{\delta}_k^*$. Assuming the presence of a linear error term, the *direct estimation* of the bias vector $\vec{\delta}_k$ incurs independent errors $\epsilon_i^{(k)}$ at each coordinate, so forming $\Delta_k = \vec{\delta}_k \vec{\delta}_k^\mathsf{T}$ yields

$$\delta_k^*(x_i)\,\delta_k^*(x_j) = \left[\delta_k(x_i) + \epsilon_i^{(k)}\right]\left[\delta_k(x_j) + \epsilon_j^{(k)}\right]$$
$$= \delta_k(x_i)\,\delta_k(x_j) + \delta_k(x_i)\,\epsilon_j^{(k)} + \delta_k(x_j)\,\epsilon_i^{(k)} + \epsilon_i^{(k)}\,\epsilon_j^{(k)}, \tag{14}$$

and therefore errors "multiply" and can amplify. By contrast, *quadratic estimation* directly predicts each entry of $\Delta_k$, so $\left(\Delta_k^*\right)_{ij} = \delta_k(x_i)\,\delta_k(x_j) + \epsilon_{ij}^{(k)}$ with only a single error term $\epsilon_{ij}^{(k)}$. This single-term error structure is much more stable in downstream computations (e.g. eigendecompositions).

## 3.3 BATCHING VIA EIGENDECOMPOSITION

Using the new cobias–covariance relationship, the term $\mathcal{L}_k$ can be written as the trace of $\Omega^{(k)}$, which can then further be expressed as the sum over the eigenvalues for the matrices constituting the cobias–covariance relationship in Equation (11), specifically

$$\mathcal{L}_k = \frac{1}{n}\sum_{i=1}^n \tau_k(x_i) = \frac{1}{n}\underbrace{\mathrm{tr}(\Omega^{(k)})}_{=\sum_i \lambda_i(\Omega^{(k)})} = \frac{1}{n}\left(\underbrace{\mathrm{tr}(\Sigma_{F_k})}_{=\sum_i \lambda_i(\Sigma_{F_k})} + \underbrace{\mathrm{tr}(\Delta_k)}_{=\vec{\delta}_k^\mathsf{T}\vec{\delta}_k} + \underbrace{\mathrm{tr}(\Sigma_Y)}_{=\sum_i \lambda_i(\Sigma_Y)}\right). \tag{15}$$

Because all of the matrices in (11) are symmetric, their eigenvalues are real with orthogonal, real eigenvectors. Moreover, since every matrix on the right hand side of Equation (11) is positive semi-definite, then $\Omega^{(k)}$ is positive semi-definite too, i.e., all eigenvalues are greater than or equal to zero. Therefore, we can write $\Omega^{(k)}$ via the eigendecomposition

$$\Omega^{(k)} = V A V^{-1}, \quad A = \mathrm{diag}(\lambda_1, \cdots, \lambda_{n'}), \quad V = [\, \vec{v}_1, \cdots \vec{v}_{n'}\,], \tag{16}$$

for $\lambda_1 \geq \lambda_2 \geq \cdots \geq \lambda_{n'}$. To choose $m$ points in one batch, for each of the $m$ largest eigenvalues $\{\lambda_j\}_{j=1}^m$ we pick the index

$$i_j \;=\; \arg \max_{1 \leq i \leq n} \big|v_j(i)\big|, \tag{17}$$

from the corresponding eigenvectors $\{\vec{v}_j\}_{j=1}^m$ and query the corresponding $x_{i_j}$. This ensures that each selected point aligns with a principal directions of $\Omega^{(k)}$ — that is, a mode that contributes the greatest variance contributions (largest eigenvalues) to the total EMSE. For the difference-PEMSE strategy, we consider the eigendecomposition of $\Omega^{(k-1)} - \Omega^{(k)}$ and select eigenvectors corresponding to the largest *positive* eigenvalues.

## 4 NUMERICAL EXPERIMENTS

### 4.1 BIAS ESTIMATION RETAINS STRONG PERFORMANCE

Because use of difference-based methods performs largely similarly to non-difference-methods for single-point acquisition (see Figure 2), we wanted to assess whether we could use either *direct estimation* or *quadratic estimation* to implement our method; full details in Appendix B. In Figure 4, we compare our method using different estimation approaches.

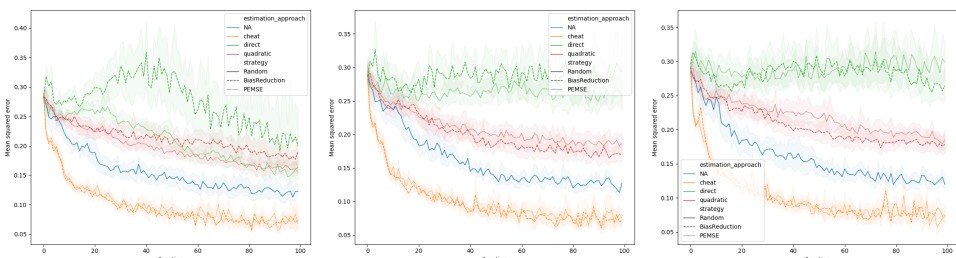

Figure 4: Assessment of BR and PEMSE acquisition function using different methods at to estimate the cobias matrix. We show all three problems in Table 2: Type I (left), Type II (middle), and Type III (right).

In Figure 4, we see that for Type I and Type II problems, either using *direct estimation* or *quadratic estimation* of the bias leads to retaining *some* of the performance gains seen when cheating (i.e., having direct access to the data distribution $Y$), but beating random performance only appears possible when sufficient initial data is available (see Appendix A). When we initialize with 10 points, the *quadratic estimation* approach is markedly better than *direct estimation*, yet the performance of *quadratic estimation* is still inferior to random selection due to the small number of initial points.

### 4.2 EIGENDECOMPOSITION AND DIFFERENCE METHODS PROVIDE BENEFIT IN SPECIFIC SCENARIOS

Up until this point, Type III problems have not been meaningfully evaluated as the correlations between draws can only be exploited in a batched setting. Now that we have a mechanism for estimating the bias, we consider the problem of selecting batches of queries; full details in Appendix B. In Figure 5, we consider random selection, when compared to selecting the top $m$ points by the PEMSE acquisition function, and then the eigendecomposition approach described in Section 3.3 using different estimation methods. We both consider using PEMSE, but also the difference-PEMSE method after the first 2 iterations (we need 2 rounds evaluated before we can use difference-PEMSE).

From Figure 5, we see that for Type I problems that PEMSE outperforms difference-PEMSE, which makes sense as there is no aleatoric noise term to cancel out in Equation (9) — so we are only adding

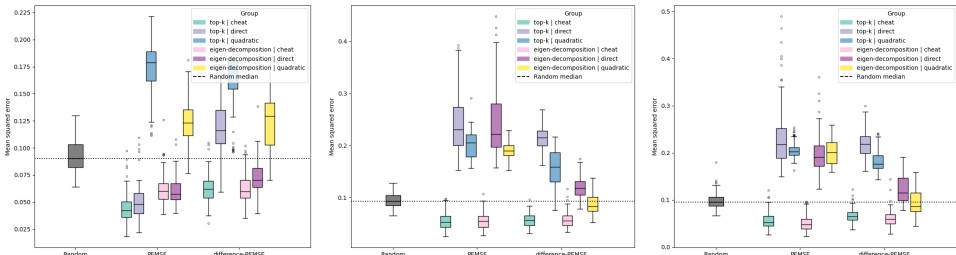

Figure 5: Assessment of PEMSE and difference-PEMSE acquisition functions using different methods at to estimate the cobias matrix. We show the performance across all 10 ensembles in the final 10 iterations. Random selection is shown as a horizontal black dotted line for comparison. We show all three problems in Table 2: Type I (left), Type II (middle), and Type III (right).

extra difficulty to our problem by requiring one to estimate the difference in the PEMSE between rounds (i.e., requiring estimation of $\Delta_k$ twice).

When considering Type II and Type III problems, PEMSE and difference-PEMSE perform similarly. Typically, *quadratic estimation* outperforms *direct estimation* when initialized with 10 initial data points, but vice versa when 100 data points are available. We also see reproducible small benefits when using the eigendecomposition approach when compared to top-$k$ selection. However, in both Type II and Type III scenarios, we see a clear benefit to using *quadratic estimation* with difference-PEMSE for the eigendecomposition approach because *quadratic estimation* will achieve higher accuracy predicting off-diagonal elements of $\Delta_k$ (stablizing the eigendecomposition) when compared to direct estimation that predicts $\vec{\delta}_k$ and then calculates $\Delta_k = \vec{\delta}_k \vec{\delta}_k^{\intercal}$ afterwards.

## 5 DISCUSSION

We have presented a novel approach to active learning leveraging the bias–variance tradeoff and integrating models across multiple rounds of experiments. This appears to contrast with Bayesian-first approaches, but these methods may be combined in due course. Moreover, our approach demonstrates the shortcomings of LC and BALD that do not appear effective in the noisy systems that we study in this work, see Figure 2.

Crucially, in the low initial data regime (initializing with 10 points), the only non-cheating method that can beat random selection in a batch setting is *quadratic estimation* with difference-PEMSE with eigendecomposition, see Figure 5. Our presented implementation leaves many avenues for further optimization. For example, we estimate unseen biases leveraging historical realizations, but we do not account for the variation in how many times a specific region of the state space is sampled; specifically, our bias estimator should have appropriately weighted training data. Using more sophisticated and stable methods for the *quadratic estimation* step should lead to a method that can be employed in all Type II and Type III problems.

There are a number of groups studying the bias–variance tradeoff in different contexts. In particular, by viewing the Kullback–Leibler (KL) divergence and mean squared error as special cases of a more general Bregman divergence (Pfau, 2013; Adlam et al., 2022). It would not be challenging to apply our approach to other Bregman divergences, yet we may not be able use the cobias–covariance approach as it is not clear how our approach would work for non-symmetric divergences, see Appendix D for more details.

In the future, we plan to investigate scenarios where the underlying system exhibits more complex behaviours. We can also construct our predictive distribution to belong to certain function classes, and as such we may not wish to sample points in a manner that leads to indescribable inferences outside the function class. For example, imagine $\mu_Y(x)$ exhibits highly complex behaviour, but we are operating in a comparatively computationally limited environment and as such $F$ belongs to a simpler class of functions; it would not make sense to resolve such intricate behaviors and we should account for this.

**Reproducibility Statement.** We have taken several steps to make our results reproducible. The problem setup, acquisition functions, and all algorithmic components are specified in §2–3, including the precise definitions of Type I/II/III tasks and noise models (Table 2, §2.3.2), the PEMSE/bias objectives (§2.2), the quadratic cobias–covariance estimator (§3.2) and batching via eigendecomposition (§3.3). We provide implementation details needed to re-run experiments in Appendix B. Source code has been supplied as a zip in the supplementary materials to reproduce our results along with documentation to run auxilliary or other bespoke scenarios.

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

# A  FURTHER NUMERICAL RESULTS WITH 100 INITIAL POINTS

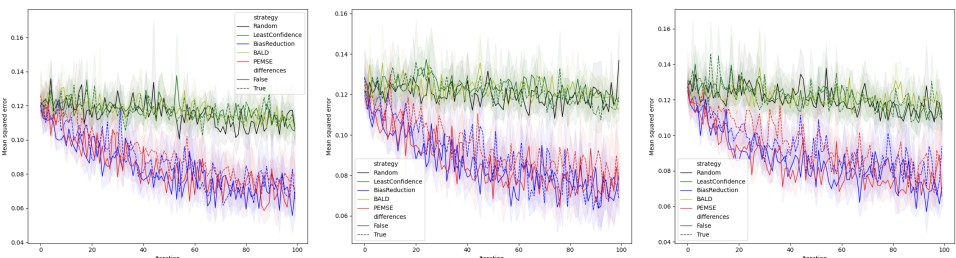

Figure 6: Repeat of Figure 2 with 100 initial points.

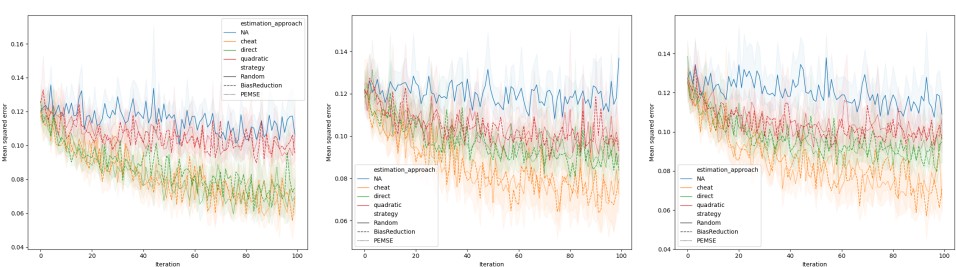

Figure 7: Repeat of Figure 4 with 100 initial points.

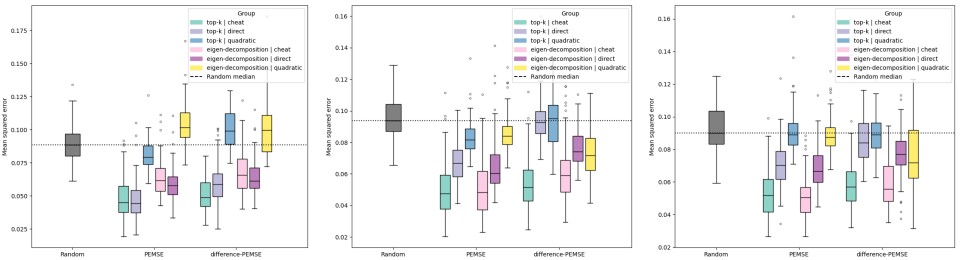

Figure 8: Repeat of Figure 5 with 100 initial points.

# B  FURTHER DETAILS OF NUMERICAL RESULTS

For the purposes of evaluating different strategies, we do not evaluate $\mathcal{L}_k$, but the key performance metric of the MSE on the unseen ground truth signal (except in Type I no-noise scenarios). We randomly initiate $\vec{x}$ as 10 or 100 random points in $\mathcal{X}$ uniformly selected using seeded instantiations to ensure comparability between strategies at initialisation. For all simulations, we discretize $x_1$ and $x_2$ into 50 points, leading $x$ to be specified by a 2500 point grid. Evaluation of our metric follows pool based sampling active learning where we calculate the MSE over the entire 2500 point space, and look to rapidly attain good inference of the function over its domain. In single acquisition experiments we observed performance over 50 iterations. In batch experiments we looked at a budget of 10 iterations querying 10 observations at a time. Each evaluation is repeated 10 times with a different set of initially labelled points. All experiments can be replicated using the supplementary code provided.

In our numerical experiments all strategies aim to improve the same model $F_k$ instantiated through an ensemble of deep neural networks. This ensemble contains 5 neural networks with 3 layers of shape (32, 32, 16) utilizing rectified linear unit activation functions (ReLU). Each neural network in the ensemble is trained via gradient descent using an Adam optimiser (Kingma & Ba, 2014) on 5 folds of the labelled training set — in our presented case study each neural network is trained on 80% of

data that is available, thereby emulating a simple form of bagging. The initial learning rate for Adam is set to 0.001, with exponential decay $\beta_1 = 0.9$ and $\beta_2 = 0.999$. Training is run until the loss does not decrease more than 0.0001 for more than 10 epochs or a maximum of 500 epochs. This simple $F_k$ is used across all experiments. Models for the task are trained with noisy labels obtained from the oracles exhibiting the noise applied to the true signal as shown in Table 2 for Type I/II/III scenarios.

Over the course of our experiments we consider strategies:

1. Strategies with perfect information on unknown PEMSE and bias, 'cheating', to motivate theoretical results in Section 2.4.

2. Strategies which use *direct estimation* of the PEMSE and bias via a Gaussian process.

3. Strategies which use *quadratic estimation* of the PEMSE and bias via symmetric neural network for matrix completion.

(1) In the first case of perfect information or 'cheating', we call upon the oracle to provide realisations $y \sim Y(x)$ which we use to impute the PEMSE or bias respectively over unknown observations. For Type I problems, realisations $y \sim Y(x)$ are without noise and hence return the true signal once. In Type II and Type III problems we call upon a noisy oracle to provide uncorrelated or correlated realisations of $y \sim Y(x)$. In this setting we call upon the oracle for 10 realisations $y \sim Y(x)$.

(2) In the second case of *direct estimation* of PEMSE or bias over unlabelled observations we rely on a small Gaussian process regressor (GP) trained to infer the PEMSE/bias $y$ using the posterior mean. The GP uses as input for observation $x_i$ a concatenated vector $\mathbf{h}_i = [\mathbf{x_i}||\mu_{F_k}(x_i)||\sigma^2_{F_k}(x_i)] \in \mathbb{R}^{d+2}$ containing the input features $\mathbf{x}_i$ which has a $d = 2$ dimensionality in our experiments, the mean prediction $\mu_{F_k}(x_i)$, and variance $\sigma^2_{F_k}(x_i)$ of the ensemble. The GP utilises a stationary anisotropic kernel implemented as a product of a constant kernel $C(1.0, [10^{-3}, 10^3])$ and an RBF kernel with initial length scaled $l_j = 1$ and bounds $[10^{-2}, 10^2]$.

$$\mathbf{k}(\mathbf{h}, \mathbf{h}^*) = \sigma_f^2 \exp\left(-\tfrac{1}{2} \sum_{j=1}^{d+2} \frac{(h_j - h_j^*)^2}{\ell_j^2}\right).$$

We add a diagonal white-noise term $\alpha = 10^{-6}$ and normalise the target PEMSE/bias to zero mean and unit variance before training. Once trained, for each *unlabelled* observation $x_i$ we compute the posterior predictive mean $\hat{y}_i$.

(3) In the third case of *quadratic estimation* we need an estimator $Q : \mathcal{X} \times \mathcal{X} \to \mathbb{R}$ to estimate $\Delta_k$ as described in Section 3.2. We use a symmetric neural neural network for matrix completion as described in Equation (13)

$$Q(x, x^*) = \psi(x)^\mathsf{T} \psi(x^*) \equiv Q(x^*, x),$$

where $\psi : \mathcal{X} \to \mathbb{R}^h$ is an input permutation invariant neural network that maps to hidden dimension of size $h = 16$. When using a neural network formulation, in order to avoid double counting off-diagonal entries, we restrict the training data to the lower triangle of symmetric matrix $\Delta_k$ (analogously, one could use the upper triangle).

For $\psi : \mathcal{X} \to \mathbb{R}^h$ we use an embedding module consisting of 3 layers with hidden dimensions (64, 64, 32) using ReLU activation functions. Each intermediate layer also has a dropout ($p = 0.1$) and batch normalisation. Like in *direct estimation* the input to our model is a concatenated vector of the input features, mean and variance of the ensemble. $Q$ is trained using stochastic gradient descent with an Adam optimiser. The initial learning rate for Adam is set to 0.0003, with exponential decay $\beta_1 = 0.999$ and $\beta_2 = 0.9$. L2 weight decay rate of 0.00001 is also used for regularisation. We split the training set into 0.85/0.15 train-validation sets in order to perform validation set based early stopping with a patience of 200 epochs on a maximum of 2000 epochs of training. After training we directly impute the bias on unlabelled entries of $\Delta_k$ (and we can calculate $\Omega_k$ using the cobias–covariance decomposition), and extract the diagonal of this matrix. For batching we also use this matrix to select indices as described in our novel batching strategy (Section 3.3).

For our experiments we used a desktop computer equipped with an 8 core Intel i9-9900 processor, NVIDIA RTX3090 GPU (for training of neural networks), with 32GB of DDR4 system memory. The sequential nature of active learning experiments and our extensive set up covering Type I/II/III scenarios, 3 estimation approaches, and 10 replicates resulted in a compute time of 7.5 days for all experiments in this manuscript. This can be trivially parallelised by running multiple experiments at once across more machines.

## C  DERIVATION OF COBIAS–COVARIANCE TRADEOFF

The cobias–covariance tradeoff becomes immediately apparent by noticing the following trick to "add zero":

$$F_k(x) - Y(x) = [F_k(x) - \mu_{F_k}(x)] + [\mu_{F_k}(x) - \mu_Y(x)] - [Y(x) - \mu_Y(x)] \tag{18}$$

Thereafter, we take the product of Equation (18) with itself at another point $x^* \in \mathcal{X}$:

$$[F_k(x) - Y(x)][F_k(x^*) - Y(x^*)] =$$
$$= [(F_k(x) - \mu_{F_k}(x)) + [\mu_{F_k}(x) - \mu_Y(x)] - (Y(x) - \mu_Y(x))]$$
$$\times [(F_k(x^*) - \mu_{F_k}(x^*)) + [\mu_{F_k}(x^*) - \mu_Y(x^*] - (Y(x^*) - \mu_Y(x^*))] \tag{19}$$

Expanding the brackets, we get

$$[F_k(x) - Y(x)][F_k(x^*) - Y(x^*)]$$
$$= [F_k(x) - \mu_{F_k}(x)][F_k(x^*) - \mu_{F_k}(x^*)] \quad \text{(Recognise key term)}$$
$$+ [F_k(x) - \mu_{F_k}(x)][\mu_{F_k}(x^*) - \mu_Y(x^*)]$$
$$- [F_k(x) - \mu_{F_k}(x)][Y(x^*) - \mu_Y(x^*)]$$
$$+ [\mu_{F_k}(x) - \mu_Y(x)][F_k(x^*) - \mu_{F_k}(x^*)]$$
$$+ [\mu_{F_k}(x) - \mu_Y(x)][\mu_{F_k}(x^*) - \mu_Y(x^*)] \quad \text{(Recognise key term)}$$
$$- [\mu_{F_k}(x) - \mu_Y(x)][Y(x^*) - \mu_Y(x^*)]$$
$$- [Y(x) - \mu_Y(x)][F_k(x^*) - \mu_{F_k}(x^*)]$$
$$- [Y(x) - \mu_Y(x)][\mu_{F_k}(x^*) - \mu_Y(x^*)]$$
$$+ [Y(x) - \mu_Y(x)][Y(x^*) - \mu_Y(x^*)] \quad \text{(Recognise key term)} \tag{20}$$

After taking the expectation over $\mathcal{F} \times \mathcal{Y}$, the three terms marked in Equation (20) will become the right hand side to Equation (11). All remaining terms become zero as

$$\mathbb{E}_{\mathcal{F}}[F_k(x) - \mu_{F_k}(x)] = \mathbb{E}_{\mathcal{F}}[F_k(x)] - \mu_{F_k}(x) = \mu_{F_k}(x) - \mu_{F_k}(x) = 0 \tag{21}$$

and analogously when taking the expectation over $\mathcal{Y}$.

## D  THE BIAS–VARIANCE TRADEOFF THROUGH THE LENS OF BREGMAN DIVERGENCES

If $\mathcal{X}$ is a closed, convex subset of $\mathbb{R}^d$, the function $D_f : \mathcal{X} \times \mathcal{X} \to \mathbb{R}$ is a Bregman divergence if there exists a strictly convex, differentiable function $f$ such that

$$D_f[\vec{u} \parallel \vec{v}] := f(\vec{u}) - f(\vec{v}) - \langle \nabla f(\vec{v}), \vec{u} - \vec{v} \rangle . \tag{22}$$

For such a Bregman divergence with arguments now replaced by random variables, we write that the average loss can be decomposed into three terms via a generalised bias–variance tradeoff (Pfau, 2013)

$$\mathbb{E}_{\mathcal{F} \times \mathcal{Y}} D_f[\vec{Y} \parallel \vec{F}] = \mathbb{E}_{\mathcal{Y}} D_f[\vec{Y} \parallel \mathbb{E}_{\mathcal{Y}} \vec{Y}] + D_f[\mathbb{E}_{\mathcal{Y}} \vec{Y} \parallel \mathcal{E}_{\mathcal{F}} \vec{F}] + \mathbb{E}_{\mathcal{F}} D_f[\mathcal{E}_{\mathcal{F}} \vec{F} \parallel \vec{F}], \tag{23}$$

where $\mathcal{E}_{\mathcal{X}} \vec{X}$ is defined as the *dual mean* for random variable $\vec{X} \in \mathcal{X}$, the primal form of the mean of $\vec{X}$ taken in dual space: $\mathcal{E}_{\mathcal{X}} \vec{X} := \text{argmin}_{\vec{z}} \mathbb{E}_{\mathcal{X}} D_f[\vec{z} \parallel \vec{X}]$. When $D_f$ is the squared Euclidean distance, Equation (23) reduces to Equation (5).

Of particular interest is when $f(\vec{v}) = \sum_i v_i \log v_i$ and thus $D_f[\vec{u} \parallel \vec{v}] = \sum_i v_i \log(u_i/v_i)$ is the Kullback-Leibler divergence over the probability simplex $\mathcal{X} = \{\vec{v} \in \mathbb{R}^d : \sum_i v_i = 1\}$. In such cases, $\mathcal{E}\vec{F} = \text{softmax}(\mathbb{E}_{\mathcal{F}} \log \vec{F})$ and the method as outlined in Section 2.2 can be used. For further reading on the topic and associated derivations, see key references (Pfau, 2013; Gruber & Buettner, 2022; Adlam et al., 2022; Yang et al., 2020; Heskes, 2025).

# E  CALCULATION OF $\omega_k(x, x^*)$ FROM DATA

Whilst these parts did not enter the final manuscript, for various numerical experiments we calculated estimates for $\omega_k(x, x^*)$ that we detail below for completeness, although in practice we used the calculation in Appendix E.3.

## E.1  UNCORRELATED DRAWS OF $Y$ ACROSS $\mathcal{X}$

For Type I and Type II problems, we are making assumption that the associated underlying probability density function for $[F_k(x), F_k(x^*), Y(x), Y(x^*)]$ factorizes, specifically

$$\rho_{\mathcal{F}^2 \times \mathcal{Y}^2}(f, f^*, y, y^*) = \rho_{\mathcal{F}^2}(f, f^*)\rho_{\mathcal{Y}}(y), \rho_{\mathcal{Y}}(y^*) \,. \tag{24}$$

For an arbitrary probability density function $\rho = \rho(z)$ with observed data points $\{z_i\}_{i=1}^N$, we can approximate $\rho$ as

$$\rho(z) \approx \frac{1}{N} \sum_{i=1}^{N} \delta(z - z_i) \,, \tag{25}$$

where $\delta$ is the Dirac delta function. We model $F_k$ as a deep ensemble with $K$ functions. Therefore, using the approximation for our factorized probability density function, we find

$$\omega_k(x_i, x_j) = \mathbb{E}_{\mathcal{F} \times \mathcal{Y}} \left\{ [F_k(x_i) - Y(x_i)] [F_k(x_j) - Y(x_j)] \right\} \tag{26}$$

$$\approx \frac{1}{KN_iN_j} \sum_{k=1}^{K} \sum_{r=1}^{N_i} \sum_{s=1}^{N_j} \left\{ [f_k(x_i) - y_r(x_i)] [f_k(x_j) - y_s(x_j)] \right\} \,, \tag{27}$$

where $N_i$ is the total number of times $Y(x_i)$ has been realised. When $N_i = 0$ or $N_j = 0$, then we cannot calculate $\omega_k$ and we have to resort to using predicted values. For the diagonal entries of $\Omega^{(k)}$, our sum simplifies to

$$\omega_k(x_i, x_i) = \tau_k(x_i) = \frac{1}{KN_i} \sum_{k=1}^{K} \sum_{r=1}^{N_i} \left\{ [f_k(x_i) - y_r(x_i)]^2 \right\} \tag{28}$$

## E.2  CORRELATED DRAWS OF $Y$ ACROSS $\mathcal{X}$

For Type III problems, our probability density function for $[F_k(x), F_k(x^*), Y(x), Y(x^*)]$ only factorises across $F$ and $Y$, in particular

$$\rho_{\mathcal{F}^2 \times \mathcal{Y}^2}(f, f^*, y, y^*) = \rho_{\mathcal{F}^2}(f, f^*)\rho_{\mathcal{Y}^2}(y, y^*) \,. \tag{29}$$

Realisations of $Y$ now have the potential to be correlated across $(x, x^*) \in \mathcal{X} \times \mathcal{X}$. To detect such correlations, we then have to specify that realisations of $Y$ are drawn together, as in, realisations in $Y$ are batched together. Therefore, the associated calculation of $\omega_k(x, x^*)$ from data becomes

$$\omega_k(x_i, x_j) = \mathbb{E}_{\mathcal{F} \times \mathcal{Y}} \left\{ [F_k(x_i) - Y(x_i)] [F_k(x_j) - Y(x_j)] \right\} \tag{30}$$

$$\approx \frac{1}{KN_{ij}} \sum_{k=1}^{K} \sum_{r=1}^{N_{ij}} \left\{ [f_k(x_i) - y_r(x_i)] [f_k(x_j) - y_r(x_j)] \right\} \,, \tag{31}$$

where index $r$ now iterates over every round of $k$ where both $Y(x_i)$ and $Y(x_j)$ were realised together and we observed this pair of realisations $N_{ij}$ times.

## E.3  BIAS-FIRST CALCULATION OF $\omega_k(x, x^*)$

Restating Equation (4), we do not have have to make any distinctions relating to the type of problem we are dealing with and therefore

$$\delta_k(x) = \mu_{F_k}(x) - \mu_Y(x) = \mathbb{E}_{\mathcal{F}}[F_k(x)] - \mathbb{E}_{\mathcal{Y}}[Y(x)]$$

$$\approx \frac{1}{K} \sum_{k=1}^{K} f_k(x_i) - \frac{1}{N_i} \sum_{r=1}^{N_i} y_r(x_i) \,. \tag{32}$$

## F  STATEMENT OF USE OF LARGE LANGUAGE MODELS

We have used OpenAI's ChatGPT model (ChatGPT 4 and ChatGPT 5) as a general assist tool to polish the writing in this manuscript.

