# OpenReview forum: "When three experiments are better than two: Avoiding intractable correlated aleatoric uncertainty by leveraging a novel bias--variance tradeoff"
_ICLR.cc/2026/Conference — Submitted to ICLR 2026_

### Official Review · Reviewer_SFiS · 2025-10-20

**Soundness:** 1
**Presentation:** 2
**Contribution:** 1
**Rating:** 2
**Confidence:** 4

**Summary:**

This paper proposes a novel framework for active learning that leverages the bias–variance–noise decomposition to better distinguish between reducible and irreducible sources of uncertainty. It introduces acquisition strategies that selectively target epistemic uncertainty, model bias, or a combination of both, enabling more principled experiment selection. They evaluate their methods on a set of synthetic toy experiments designed with controlled noise assumptions, showing that their approach outperforms standard uncertainty-based baselines such as BALD and LC.

**Strengths:**

1. The paper is well written.

2. The separation of uncertainties through the bias–variance–noise decomposition is intuitive and well-motivated, making the proposed framework conceptually clear and broadly applicable across different active learning settings.

**Weaknesses:**

1. The paper does not consider or compare against recent state-of-the-art active learning methods such as BatchBALD [1], BADGE [2], and BAIT [3]. These approaches are well-established benchmarks for active learning.

2. The paper relies entirely on toy experiments to demonstrate its effectiveness. While these experiments are useful for illustrating theoretical properties, the method is not purely theoretical and would therefore benefit significantly from evaluation on real-world datasets.

3. The structure of the paper is somewhat confusing. Section 2 includes a numerical experiment comparing the proposed methods to BALD and showing improved performance, yet BALD is subsequently disregarded in the main numerical experiment section, Section 4.

4. It is not explicitly stated what models are being trained in the experiments. Are they are neural networks, Gaussian processes, random forests, or linear models?

5. There is no limitations or weakness sections.

6. Minor Issues:

- The caption for Figure 1 blends into the body text and should be visually separated for improved readability.
- A visualization of each dataset listed in Table 2 would make the paper more accessible and easier to interpret.
- On line 157, should the expression be $\delta_{\inf}(x) = 0$ instead of $\delta_{\inf} = 0$?
- The legends in Figures 2 and 4 are confusing. For example, in Figure 4 the solid blue line is unlabeled, and the legend references a “random” (solid black) line that does not appear in the figure. Consider moving legends outside the plot and ensuring every line has a corresponding label.

[1] Kirsch, A., van Amersfoort, J., & Gal, Y. (2019). BatchBALD: Efficient and Diverse Batch Acquisition for Deep Bayesian Active Learning. NeurIPS 2019.

[2] Ash, J. T., Zhang, C., Krishnamurthy, A., Langford, J., & Agarwal, A. (2020). Deep Batch Active Learning by Diverse, Uncertain Gradient Lower Bounds (BADGE). ICLR 2020.

[3] Kim, J., Lee, Y., & Shin, J. (2021). BAIT: Bayesian Active Learning by Information Theoretic Measures. NeurIPS 2021.

**Questions:**

1. Why is Random better than BALD? Figure 2

2. Why does Figure 4 & 5 include no BALD or other baseline methods?

---

> ### Author Response · Authors · 2025-11-23
> **Response to SFiS (Part 1 of 2)**
>
> We thank the reviewer for their comments and appreciate that you found our bias-variance-noise decomposition framework clear and intuitive. Below we address your specific questions and return to weaknesses.
>
> *"Q1 Why is Random better than BALD? Figure 2"*
>
> Random acquisition has been shown to be a powerful baseline to beat in most AL scenarios even with perfect information as we have shown in Figure 2. There is no guarantee that BALD should perform better than random for all problems.
>
> *"W1. The paper does not consider or compare against recent state-of-the-art active learning methods such as BatchBALD [1], BADGE [2], and BAIT [3]. These approaches are well-established benchmarks for active learning."*
>
>
> We agree partially. we have included BALD in our single acquisition upon which BatchBALD obviously builds. However, in considering its addition we find that it is primarily defined for classification problems [1]; not the regression problems we wish to tackle. Would you agree that the regression definition by Holzmüller et al. [A] is suitable? Naturally, we would have to adapt and translate it for our task model of a deep ensemble.
>
> [A] David Holzmüller, Viktor Zaverkin, Johannes Kästner, and Ingo Steinwart. 2023. A framework and benchmark for deep batch active learning for regression. J. Mach. Learn. Res. 24, 1, Article 164 (January 2023), 81 pages.
>
> *"W2: The paper relies entirely on toy experiments to demonstrate its effectiveness. While these experiments are useful for illustrating theoretical properties, the method is not purely theoretical and would therefore benefit significantly from evaluation on real-world datasets."*
>
> We agree that real world experiments would be great, but disagree with the premise that they are a prerequisite to this manuscript. The current setup was deliberately chosen for control, reproducibility, and noise diversity of exploring active learning scenarios which allow for infinite measurements of the same observation under our Type I, Type II, Type III scenarios. Doing so realistically would inevitably result in the use of a simulator as we have done. No existing static dataset would allow for this. If you believe it would be convincing we can add heteroscedastic noisy oracles on top of existing datasets.
>
> Performing the Type III structured noise experiments would have required the running of active learning scenarios potentially measuring the same observations across multiple strategies and iterations. If we had used perturb-seq experiments as we described in the introduction this would be prohibitively expensive; and we would not be able to guarantee a good representation of batch or technical noise. Even more, challenging would be the fact that no review would independently be able to reproduce results. Thus the best option to study our contributions under the desired scenarios would be such in-silico simulators. This could be an interesting future contribution to set up more simulators to study these real active learning scenarios and would be key to their adoption in realistic scenarios.
>
> *"W3: The structure of the paper is somewhat confusing. Section 2 includes a numerical experiment comparing the proposed methods to BALD and showing improved performance, yet BALD is subsequently disregarded in the main numerical experiment section, Section 4."*
>
> This is to focus the comparison of the proposed methods in its variations under direct estimation of PEMSE/Bias, its use of difference based operator (Equation 8); both in single and batch acquisition (where we further explore Top-K versus eigen-decomposition based batch acquisition). This was after the proposed methods are compared to existing methods under the ‘cheating’ scenario figure which you find in Section 2 and found the proposed to outperform. We could put the other existing methods in for comparison but find that this can clutter the paper significantly under the space limits of the conference. We can add them if it helps.
>
> *"W4. It is not explicitly stated what models are being trained in the experiments. Are they are neural networks, Gaussian processes, random forests, or linear models?"*
>
> To perform the regression task we are using a simple MLP fully described in Appendix B, alongside our experimental design.

---

> ### Author Response · Authors · 2025-11-23
> **Response to SFiS (Part 2 of 2)**
>
> *"W5. There is no limitations or weakness sections."*
>
> We were not aware that this is a requirement under the Author Guide as in some other venues. However, we agree it is important to be open about these; mostly limited by space in the main text we have described some in our appendix. To summarise, our approach as a number of key limitations. First, the estimation of the bias is challenging, and in some cases the quadratic approach does not offer meaningful benefits. Due to the stochastic gradient decent and the random nature of training neural networks, the difference-based methods may not be optimal outside of batched settings and is worthy of further investigation. Finally, our work shown for regression problems and it is not clear how well this approach will work for classification problems with non-symmetric cobias-covariance tradeoffs, see Appendix C.
>
> "*W6. Minor Issues:
> The caption for Figure 1 blends into the body text and should be visually separated for improved readability.
> A visualization of each dataset listed in Table 2 would make the paper more accessible and easier to interpret.
> On line 157, should the expression be  instead of ?
> The legends in Figures 2 and 4 are confusing. For example, in Figure 4 the solid blue line is unlabeled, and the legend references a “random” (solid black) line that does not appear in the figure. Consider moving legends outside the plot and ensuring every line has a corresponding label.*"
>
> Thank you very much for these suggestions!

---

> > ### Comment · Reviewer_SFiS · 2025-11-25
> >
> > I would like to thank the authors for their comments. However, I still believe that the manuscript lacks adequate baselines. The paper you cite [1] is focused on regression tasks and incorporates BADGE, BAIT, and BatchBALD as baselines. Additionally, I would like to point out an alternative source that also focuses on regression tasks with the established baselines [2].
> >
> > In my opinion, the experiments presented in the manuscript are still too simplistic. The figures use RMSE as a metric to differentiate between active learning methods, similar to [2], but I question why active baselines are not considered in the context of robotic dynamics. This would represent a more practical use case, as collecting robotic data is expensive.
> >
> > For these reasons, I maintain my score.
> >
> > [1] David Holzmüller, Viktor Zaverkin, Johannes Kästner, and Ingo Steinwart. 2023. A framework and benchmark for deep batch active learning for regression. J. Mach. Learn. Res. 24, 1, Article 164 (January 2023), 81 pages.
> >
> > [2] Berry, Lucas, and David Meger. "Epistemic Uncertainty Estimation in Regression Ensemble Models with Pairwise Epistemic Estimators." The Thirty-ninth Annual Conference on Neural Information Processing Systems.

---

### Official Review · Reviewer_pAea · 2025-10-24

**Soundness:** 4
**Presentation:** 4
**Contribution:** 3
**Rating:** 6
**Confidence:** 3

**Summary:**

The authors propose a novel quantity derived from the bias-variance trade-off of statistical models that can serve as an optimization target for Active Learning (AL) acquisition functions.
They explain the derivation of this quantity in great detail and construct a series of toy experiments with different structures in the noise term to showcase the properties of their proposed method.
As this novel quantity is unknown in practice, the authors propose several way of approximating it and benchmark the variants of their methods against existing AL methods, as well as "cheated" version of their method that has direct access to the target quantity.

**Strengths:**

- Very well structured construction of the novel quantity that the authors are planning to optimize
- Principled experiment to show the potential of they method (if a perfect approximation would be possible)
- Multiple approaches to approximate the novel quantity and initial comparison against competitors
- Clear path for improving upon the proposed method

**Weaknesses:**

- (Minor) It is well known in AL that uncertainty-based methods (like the proposed one) are struggling for small labeled sets, as the model F cannot be appropiately trained. A comparison of Fig. 2 and Fig. 6 (App. A) clearly hints in this direction. For that reason, we really would have liked to see a comparison against a diversity based method like TypiClust or ProbCover to acertain, if the "cheat" variant of the proposed algorithm can surpass diversity based acquisition functions.
- (Minor) Your comparison between "BiasReduction" and "PEMSE" variants (Fig. 4) is lacking depth. Both methods appear to be equal in performance, which should only be true if we only use MSE as metric. The advantage the PEMSE method can generate can only be visible in the calibration of the network, which should be easy to check by comparing $\sigma_F$ to $\sigma_Y$ at all 2500 points of your data grid. This way you could highlight the advantage that PEMSE should have over BiasReduction for model training.

**Questions:**

- In Fig. 2, do you mean "Cheat" and "Approximated" instead of "False" and "True"?
- Is Fig. 4 missing a curve for Random or is "NA" supposed to be random?

---

> ### Author Response · Authors · 2025-11-23
> **Response to pAea**
>
> We thank the reviewer for their comments and recognizing the importance of the problem and the principled approach to defining the novel quantities we want to study. We appreciate the acknowledgement that we have tried to experiment and test the problem in a structured approach for scenarios with perfect information and those without. It is especially challenging, to get this correct for our scenario which requires a simulator to provide an infinitely queryable pool based problem. Below we address your specific questions and return to weaknesses.
>
> *"Q1. In Fig. 2, do you mean "Cheat" and "Approximated" instead of "False" and "True"?*"
>
> In this context, all methods have access to perfect information and are thus “cheat” methods. The False and True refers to whether our proposed methods use the difference formulation (Equation 8) or not.
>
> *"Q2. Is Fig. 4 missing a curve for Random or is "NA" supposed to be random?*"
>
> You are correct. The blue line denotes the random strategy.
>
>
> *"W1. (Minor) It is well known in AL that uncertainty-based methods (like the proposed one) are struggling for small labeled sets, as the model F cannot be appropiately trained. A comparison of Fig. 2 and Fig. 6 (App. A) clearly hints in this direction. For that reason, we really would have liked to see a comparison against a diversity based method like TypiClust or ProbCover to acertain, if the "cheat" variant of the proposed algorithm can surpass diversity based acquisition functions.*"
>
> Thank you for that suggestion, we will look into adding a diversity based method as part of the evaluation.
>
> *"W2. (Minor) Your comparison between "BiasReduction" and "PEMSE" variants (Fig. 4) is lacking depth. Both methods appear to be equal in performance, which should only be true if we only use MSE as metric. The advantage the PEMSE method can generate can only be visible in the calibration of the network, which should be easy to check by comparing  to  at all 2500 points of your data grid. This way you could highlight the advantage that PEMSE should have over BiasReduction for model training."*
>
> Thank you for that suggestion, however we are not entirely sure what is meant in calibration of the network. Is this referring to calibration curves? We are comparing the MSE computed over all 2500 points in the data grid, as a measure of the function inferring $y$ over the entire input space.

---

> > ### Comment · Reviewer_pAea · 2025-11-24
> >
> > Extension of W2:
> >
> > In your own words: "one would ideally like to select points in the state space such that the bias is close to zero, but also that
> > the predictive distribution approximately matches the underlying noisy oracle" (line 088).
> > In Figure 4 you test the proposed algorithm (PEMSE) vs a reduced form that only reduces the bias (BiasReduction).
> > The used metric is MSE, so you are only measuring performance in terms of the prediction error (bias). \
> > Since both of the approaches aim to reduce the bias, they should be nearly equivalent (as shown in the figure).
> > The real question is: "How much better does the predictive distribution of the PEMSE-trained model match the noisy oracle compared to a BiasReduction-trained model?"
> > This is commonly referred to as "model calibration" and since it poses the main contribution of your algorithm should be measured and highlighted directly.

---

### Official Review · Reviewer_FYQx · 2025-10-31

**Soundness:** 1
**Presentation:** 1
**Contribution:** 2
**Rating:** 2
**Confidence:** 3

**Summary:**

The paper presents a novel approach to active learning (AL) that seeks to optimize data acquisition by explicitly
addressing the bias-variance tradeoff in noisy, heteroskedastic and correlated real-world settings. The core idea is to
leverage a decomposition of the pointwise expected mean squared error into epistemic uncertainty, bias, and irreducible
aleatoric uncertainty. The authors introduce a "cobias-covariance tradeoff" and propose a quadratic bias estimation method,
to guide data point acquisition that effectively reduces model bias and epistemic uncertainty.

**Strengths:**

The paper offers a compelling motivation for using the bias-variance tradeoff as a guiding principle for AL acquisition
functions. The explicit consideration of heteroskedastic and correlated noise is generally an interesting and practically
relevant setting.

**Weaknesses:**

- **Major Presentation and Structural Issues:** The paper's most significant weakness lies in its poor presentation and
lack of clear structure. The narrative is disjointed, frequently mixing theoretical derivations with experimental
results and ablation studies across various sections (e.g., experimental results in Section 2.4, followed by another
dedicated Section 4). This significantly hinders readability and comprehension. A dedicated section for related work is
also notably absent, making it difficult to contextualize the contributions.
- **Lackluster Problem Statement and Notation:** The problem statement and introduction of notation (Section 2.1) are vague
and inconsistent. Here are a few examples:
  - The variable $Y$ is initially introduced as a "random variable in space $\mathcal{Y}$" but later treated as a
  stochastic process $Y(x)$
  - What does $\mathbb{E}_\mathcal{Y}[]$ mean? Is $\mathbb{Y}$ a probability space?
  - $Q_k \cap L_k = \emptyset$ (section 2.3.1) should be $Q_k \cap L_{k-1} = \emptyset$
  - Equation 7, introducing a "gradient" with respect to an integer step 'k', is mathematically ill-defined.
- **Insufficient Experimental Detail in Main Text:** Crucial information about the experimental setup, such as the specific
model architecture used is omitted from the main body. I acknowledge that Appendix B contains details on the experiments.
However, this makes it impossible for readers to fully grasp the methodology without referring to supplementary material,
which should ideally contain additional details, not core methodology.
- **Limited Experimental Scope:** The evaluation only considers three specifically designed toy examples. While these toy
examples aim to highlight specific aleatoric noise conditions (i.e. no noise, uncorrelated noise and correlated noise),
the absence of evaluation on more (realistic) benchmarks raises questions about the generalizability and practical
applicability of the proposed methods.
- **Unclear Algorithmic Variants and Baselines:** The descriptions of algorithmic variants and baselines are not consistently
clear, making it challenging to understand how comparisons were made.

Based on the significant issues in presentation, clarity, and structural organization, alongside some fundamental
inconsistencies in mathematical notation and insufficient experimental detail in the main text, I strongly recommend
rejection of this paper in its current form.

**Questions:**

No specific questions, please comment on the points below "Weaknesses"

---

> ### Author Response · Authors · 2025-11-23
> **Response to FYQx (Part 1 of 2)**
>
> We thank the reviewer for the detailed feedback and for highlighting both the novel conceptual motivation and the areas where the paper can be clarified and strengthened. In particular, we appreciate the acknowledgement that “ explicit consideration of heteroskedastic and correlated noise is generally an interesting and practically relevant setting”. However, several parts of this review appears to be in contradiction with some other reviews we’ve received. We address the comments below, but in our opinion, this review exaggerates small flaws, claiming they are major shortcomings, and unfairly characterizing the work.
>
> "*W1: Major Presentation and Structural Issues: The paper's most significant weakness lies in its poor presentation and lack of clear structure. The narrative is disjointed, frequently mixing theoretical derivations with experimental results and ablation studies across various sections (e.g., experimental results in Section 2.4, followed by another dedicated Section 4). This significantly hinders readability and comprehension. A dedicated section for related work is also notably absent, making it difficult to contextualize the contributions.*"
>
> While we appreciate the concern, this comment does not align with the structure outlined and with the comments we have received from the other reviewers. Section titles make the progression explicit: we introduce our framework and method in Section 2 “Method Concept”, motivating our work with experiments under an unrealistic (but often used!) perfect information scenario. Based on these results we focus on the adaptations we have to make to our method for it to work in practice within Section 3 “Method in practice” finally followed by Section 4 “Numerical Experiments”. Related work is sparse in our problem setting and did not warrant a dedicated and delineated section. Related and background is integrated within the introduction section
>
> "*W2. Lackluster Problem Statement and Notation: The problem statement and introduction of notation (Section 2.1) are vague and inconsistent. Here are a few examples:*"
>
> *"W2a. The variable $Y$  is initially introduced as a "random variable in space $\mathcal{Y}$" but later treated as a stochastic process $Y(x)*"
>
> $Y(x)$ is not a stochastic process in the traditional sense, $x$ is a state space variable, not a time variable, thus $Y(x)$ is a stochastic process indexed by the state space. This is used consistently throughout, and the appendix provides the formal measure-theoretic setup.
>
> *"W2b. What does $\mathbb{E}_{y}[] mean? Is \mathbb{Y} a probability space?"*
>
> $\mathcal{Y}$ is the label/output space $Y$ is the random variable in that space. For example in the experiments $\mathcal{Y}$ is real number space, and $Y$ the random variable in this space.
>
> $\mathbb{E}_{\mathcal{Y}}[W(x)]$ as in equation 3 is the expected value of the label for the noise pattern $W(x)$ (noise pattern at $x$).
>
> *"W2c. $Q_{k} \cap L_{k} = \emptyset$ (section 2.3.1) should be $Q_{k} \cap L_{k-1} = \emptyset$"*
>
> Thanks for this!
>
> *"W2d. Equation 7, introducing a "gradient" with respect to an integer step 'k', is mathematically ill-defined."*
>
> We are very clear lines 222-223 (and onwards) that we are not taking a limit for $k$, we refer to gradients to build intuition around the approximation of the change of the PEMSE. In other words, this is a finite-different operator, not a calculus derivative. We do not rely on any derivations assuming continuous operators.
>
> *"W3. Insufficient Experimental Detail in Main Text: Crucial information about the experimental setup, such as the specific model architecture used is omitted from the main body. I acknowledge that Appendix B contains details on the experiments. However, this makes it impossible for readers to fully grasp the methodology without referring to supplementary material, which should ideally contain additional details, not core methodology."*
>
> As you mention the experimental setup is written in detail within Appendix B and disagree that it is impossible for readers to grasp the methodology of the methodology without it. This is not an uncommon structure especially in the applied sciences such as bioinformatics. The model which performs the task is secondary to the AL strategies being presented and could be replaced with any other estimator to perform the same task.The reviewer acknowledges the details exist but criticises their placement, which is a matter of format, not omission.

---

> > ### Author Response · Authors · 2025-11-23
> > **Response to FYQx (Part 2 of 2)**
> >
> > *"W4 Limited Experimental Scope: The evaluation only considers three specifically designed toy examples. While these toy examples aim to highlight specific aleatoric noise conditions (i.e. no noise, uncorrelated noise and correlated noise), the absence of evaluation on more (realistic) benchmarks raises questions about the generalizability and practical applicability of the proposed methods.*"
> >
> > As described by the reviewer, we designed the simulators to explicitly exhibit the noise scenarios we wish to study in detail. We also agree that our work would be more convincing when applied to real life scenarios. However, we disagree with the premise that a ‘real’ task is a prerequisite to the contributions being considered. The current setup was deliberately chosen for control, reproducibility, and noise diversity of exploring active learning scenarios which allow for infinite measurements of the same observation under our Type I, Type II, Type III scenarios. The key word being reproducibility, and ability for independent reproduction of our experiments. No existing static dataset would allow for this.
> >
> > *"W5. Unclear Algorithmic Variants and Baselines: The descriptions of algorithmic variants and baselines are not consistently clear, making it challenging to understand how comparisons were made."*
> >
> > Please, refer to Table 1 for all algorithm variants and Figure 3 for the workflow. As for comparisons, Figure 2 compares proposed strategies and existing strategies under perfect information, as is typical in many active learning literature; but unrealistic to real life noise scenarios. Figure 4 shows numerical results when the oracle exhibits different noise profiles. Figure 5, shows the potential benefits of using an eigendecomposition based approach to constructing batches of acquisitions under the different noise profiles.

---

### Meta-Review · Area_Chair_fPci · 2026-01-01

**Summary:**

This paper proposes a collection of active learning strategies derived from a bias–variance aleatoric noise decomposition. Reviewers generally agreed that the conceptual framing around the bias–variance tradeoff is interesting. However, they also raised major concerns about the current form of the study, which the rebuttal did not fully address. First, the empirical validation is limited almost entirely to carefully constructed toy simulators; while useful for controlled analysis, this evaluation does not convincingly demonstrate practical relevance. Second, the paper lacks adequate benchmarking against established state-of-the-art active learning baselines, leaving it unclear how the proposed methods compare beyond a narrow experimental scope. Taken together, these issues suggest that the paper in its current form is not yet ready for acceptance, and therefore the AC, sadly, recommends ``rejection``.

**Reviewer Concerns:**

Reviewer **FYQx** raised concerns about the limited experimental scope, stating that the evaluation is confined to a small set of toy problems, making it difficult to assess the generality and practical relevance of the proposed methods. The reviewer also raised concerns related to clarity of presentation, paper structure, and aspects of mathematical notation (the authors provided clarifications in the rebuttal in this regard, however, the rebuttal did not resolve the primary concern regarding empirical validation).

Reviewer **SFiS** emphasized missing comparisons to established active learning baselines and questioned the reliance on synthetic and toy experiments. Again, these concerns were not substantively addressed, and the reviewer explicitly maintained their score.

Reviewer **pAea** was generally positive about the conceptual construction and theoretical motivation. However, **pAea** also highlighted missing diversity-based baselines, which remain unaddressed.

Overall, while some clarifications and minor issues were resolved, the central concerns around experimental breadth, baseline coverage, and demonstrated impact remain outstanding.

**Reviewer Scores:**

Based on the rebuttal and subsequent discussion, the AC does not expect reviewers **FYQx** or **SFiS** to increase their scores, as their core objections remain unresolved. The AC does not expect the reviewer **pAea** to actively champion acceptance. In light of the other reviews and unresolved concerns, it is likely that **pAea** would converge to a more cautious position and downgrade their score.

---

### Decision · Program_Chairs · 2026-01-26

Reject